# Layered metals as polarized transparent conductors

**Carsten Putzke** [1,2] ✉, **Chunyu Guo** [1], **Vincent Plisson**[3], **Martin Kroner**[4], **Thibault Chervy**[4,5], **Matteo Simoni**[4], **Pim Wevers** [4], **Maja D. Bachmann** [6,7], **John R. Cooper**[8], **Antony Carrington** [9], **Naoki Kikugawa** [10], **Jennifer Fowlie**[11], **Stefano Gariglio** [11], **Andrew P. Mackenzie** [6,7], **Kenneth S. Burch** [3], **Ataç İmamoğlu**[4] & **Philip J. W. Moll** [1,2] ✉

The quest to improve transparent conductors balances two key goals: increasing electrical conductivity and increasing optical transparency. To improve both simultaneously is hindered by the physical limitation that good metals with high electrical conductivity have large carrier densities that push the plasma edge into the ultra-violet range. Technological solutions reflect this trade-off, achieving the desired transparencies only by reducing the conductor thickness or carrier density at the expense of a lower conductance. Here we demonstrate that highly anisotropic crystalline conductors offer an alternative solution, avoiding this compromise by separating the directions of conduction and transmission. We demonstrate that slabs of the layered oxides $Sr_2RuO_4$ and $Tl_2Ba_2CuO_{6+\delta}$ are optically transparent even at macroscopic thicknesses >2 μm for c-axis polarized light. Underlying this observation is the fabrication of out-of-plane slabs by focused ion beam milling. This work provides a glimpse into future technologies, such as highly polarized and addressable optical screens.

Transparent conductors are key to electronic components that interact with visible light, such as displays, photovoltaics and light-emitting diodes. Creating a transparent metal requires a fine-tuning process of materials parameters in the optically visible range of the spectrum. On the one hand, one needs to ensure that interband transitions only occur above the ultra-violet to avoid absorption. On the other hand, the itinerant electrons should react slowly compared to the incoming light wave, to avoid screening and reflection. The latter is parametrized by the plasma frequency $\omega_p = \frac{e}{\epsilon_0 \epsilon_r} \sqrt{\frac{n}{m^*}}$[1], where $e$ denotes the electron charge and $\epsilon_0$ the vacuum permittivity. The dielectric constant $\epsilon_r$ charge carrier density $n$ and effective electron mass $m^*$ are material specific parameters. Pure metals such as copper, gold, and palladium possess high carrier densities and high conductivity, Fig. 1B. Consequently, those materials have a plasma frequency in the blue to ultra-violet range[2], giving rise to the metallic appearance due to high reflectivity and interband absorption. The most straightforward approach to shift the plasma frequency below the visible range is to utilize materials with a reduced $n$. This in turn leads to a reduction in electrical conductivity in the Drude formalism[1], $\sigma = e^2 \tau \frac{n}{m^*}$, with time. While tuning the carrier density is more prevalent, there have been interesting approaches to reduce $\omega_p$ via effective mass tuning in electronically correlated quantum materials[3]. However, the high

[1]Institute of Materials, École Polytechnique Fédérale de Lausanne (EPFL), 1015 Lausanne, Switzerland. [2]Max Planck Institute for the Structure and Dynamics of Matter, Hamburg 22761, Germany. [3]Department of Physics, Boston College, Chestnut Hill, MA 02467, USA. [4]Institute of Quantum Electronics, ETH Zurich, CH-8093 Zürich, Switzerland. [5]NTT Research, Inc., Physics and Informatics Laboratories, 940 Stewart Drive, Sunnyvale, CA 94085, USA. [6]Max Planck Institute for Chemical Physics of Solids, 01187 Dresden, Germany. [7]School of Physics and Astronomy, University of St Andrews, St Andrews KY16 9SS, UK. [8]Department of Physics, University of Cambridge, Madingley Road, Cambridge CB3 0HE, UK. [9]H.H. Wills Physics Laboratory, University of Bristol, Tyndall Avenue, Bristol BS8 1TL, UK. [10]National Institute for Materials Science, Ibaraki 305-0003, Japan. [11]Department of Quantum Matter Physics, University of Geneva, 1211 Geneva, Switzerland. ✉e-mail: Carsten.Putzke@mpsd.mpg.de; Philip.Moll@mpsd.mpg.de

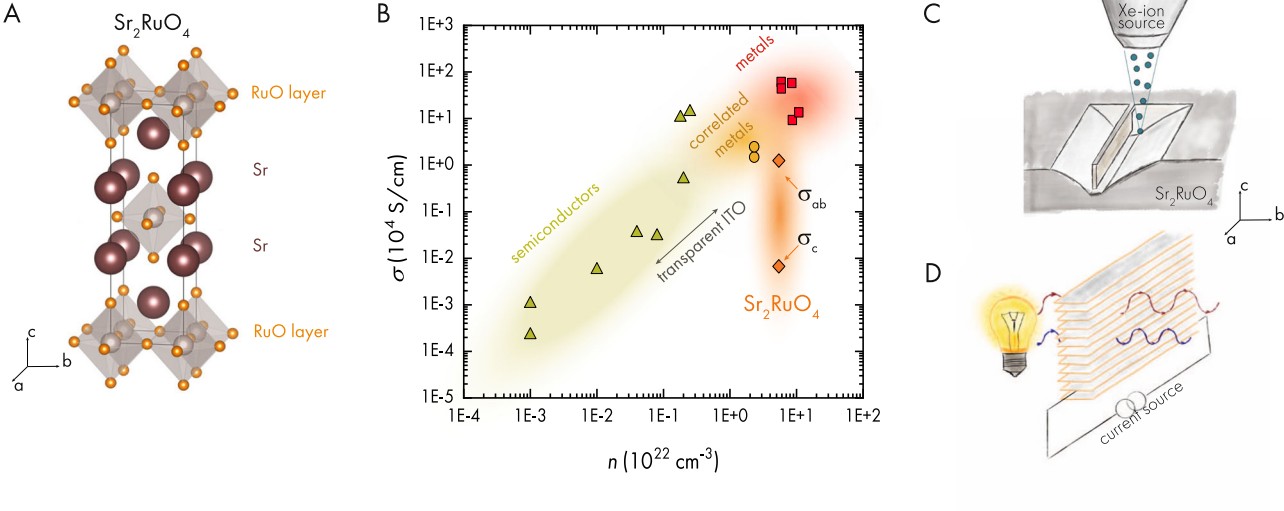

**Fig. 1 | Transparent correlated electron system Sr$_2$RuO$_4$. A** Crystal structure of SRO. Highly conductive RuO-layers are separated by Sr donor atoms. **B** Electrical conductivity versus carrier density of indium-tin-oxide (ITO, triangle)[36–41], metals (squares, Ni, Cu, Au, Pd)[42,43] and correlated metals (circles: SrVO$_3$ and CaVO$_3$ REF; diamonds: SRO)[5,6,44]. For ITO the region of transparent thin film conductors is indicated. For SRO the in-plane ($\sigma_{ab}$) and out-of-plane ($\sigma_c$) conductivity is shown, where $\frac{\sigma_{ab}}{\sigma_c} > 100$. The $c$-axis conductivity in SRO is found to be similar to that of

transparent conductors like ITO. **C** Schematic illustration of the focused ion beam micro-machining process used to obtain a slab of SRO with the shortest direction along an in-plane lattice vector. **D** Sketch of the main result found in this work. An in-plane confined ac-slab of SRO is back illuminated by a white light source while simultaneously allowing measurements of the in-plane electrical conductivity. Light in the visible range is transmitted through a macroscopic thickness of SRO.

carrier density of metals still requires thin films <100 nm in thickness for high optical transmission, significantly increasing the device resistance and thus dissipative losses through heating.

One of the most widely used transparent conductors is indium tin oxide (ITO) which presents a good compromise between conductivity and optical transmission. However, the field continues its search for alternative materials with lower cost, better mechanical properties[4], simpler processing conditions, or additional functionality, depending on the application.

## Results

Here we report a different way of obtaining transparent conductors based on the separation of conducting and transmitting directions in highly anisotropic metals. To demonstrate this generic behavior expected for strongly anisotropic conductors, we focus on the metal Sr$_2$RuO$_4$ (SRO). The crystal structure of SRO consists of RuO-layers separated by strontium, Fig. 1A. The structural anisotropy is reflected in the electrical properties of the material. At room temperature the out-of-plane conductivity is reduced by more than two orders of magnitude[5,6] as compared to the in-plane value, Fig. 1B. Nevertheless, the out-of-plane conductivity of SRO is comparable to that of ITO.

The challenge of this conceptually simple idea lies in the device fabrication. While devices that are sufficiently thin along the c-axis can be produced by exfoliation, e.g., graphene, the fabrication of a slab with thin b-axis goes against the natural growth direction and is hence more challenging. Mechanical techniques to thin down a crystal, such as polishing, inevitably induce unwanted cleaving and hence device cracking. The focused ion beam (FIB) is a kinetic micro-machining method and exerts only negligible force onto the sample, and thus is ideal to carve out-of-plane slabs out of bulk single crystals[7,8]. Those slabs are fabricated such that the shortest edge is along the in-plane direction, Fig. 1C. In this configuration we can access the in-plane electrical properties of SRO while at the same time investigating the optical transmission along the quasi-2D sheets, b-axis, Fig. 1D.

The resulting experimental setup is shown in Fig. 2A. A 400-nm-thick slab of SRO was mounted on a sapphire substrate with pre-patterned Al-contacts by using in situ FIB-assisted platinum

deposition. The sample was then structured using a Xenon-plasma FIB to allow for a four-point in-plane resistivity measurement, Fig. 2B. The room temperature resistivity is found to be unaltered from previous reports in the bulk[6], suggesting negligible introduction of defects when machining, in agreement with the optical transparency. Our SRO slabs remain significantly larger than the room temperature mean free path thereby preserving the bulk resistivity. In thin films of ordinary metals the room temperature resistivity is enhanced over the bulk resistivity due to finite size effects[9–12] making it comparable to that of SRO. To avoid cleaving by differential thermal contraction, a suspended mounting method offering low stress was developed for low temperature measurements (see "Methods"). In the low temperature regime bulk single crystals of SRO show resistivity values of <1 μΩcm[6], while our device remained more resistive. Given the long mean-free-path of these clean crystals at low-temperatures, a finite size enhancement of scattering due to confinement is not unexpected[12]. The in-plane electrical conductivity of SRO at room temperature was determined to be 1.25 S/cm. The electrical anisotropy of bulk single crystals can be estimated at 200, which agrees well with the room temperature anisotropy in Fig. 2A. The c-axis conductivity results in 7 mS/cm which is in good agreement with the observed transparency at macroscopic thickness.

The optical conductivity of SRO for linearly polarized light is shown in Fig. 2C. The mechanism for transparency is based on separating the directions of high and low electrical conductivity via the crystalline anisotropy of SRO. As a result, one naturally expects the transmission to strongly depend on the polarization of the light. Linearly polarized light with an electric field in the in-plane direction is readily screened by the highly mobile in-plane electrons and hence reflected. Out-of-plane electric fields, on the other hand, are barely screened and such photons will be transmitted over long distances. To illustrate this effect experimentally, we prepared two SRO slabs of 1 μm thickness along the b-axis (Fig. 3). The slabs were placed on a non-transparent Al film on a sapphire substrate with prepatterned holes (EPFL logo). The samples were studied under a bright-field microscope equipped with transillumination and a linear polarization filter. Using a micro-manipulator, the first sample was placed in direct contact with

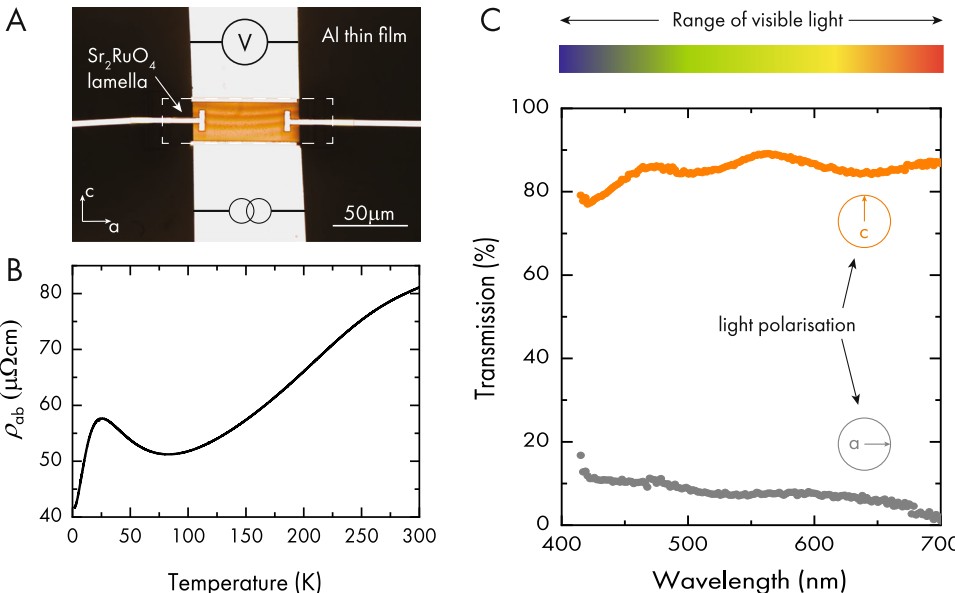

**Fig. 2 | 400 nm thick lamella. A** An optical image of the lamella on sapphire. The resistivity measurement is schematically shown. Resistivity of Sr$_2$RuO$_4$ with in-plane confinement (**B**). Transmission in the range of visible light is shown in (**C**). Two different polarization directions were measured. One with linearly polarized light along the planes and one with the light linearly polarized perpendicular to the planes. While in-plane light is mostly absorbed, light polarized out-of-plane is transmitted with more than 80% efficiency.

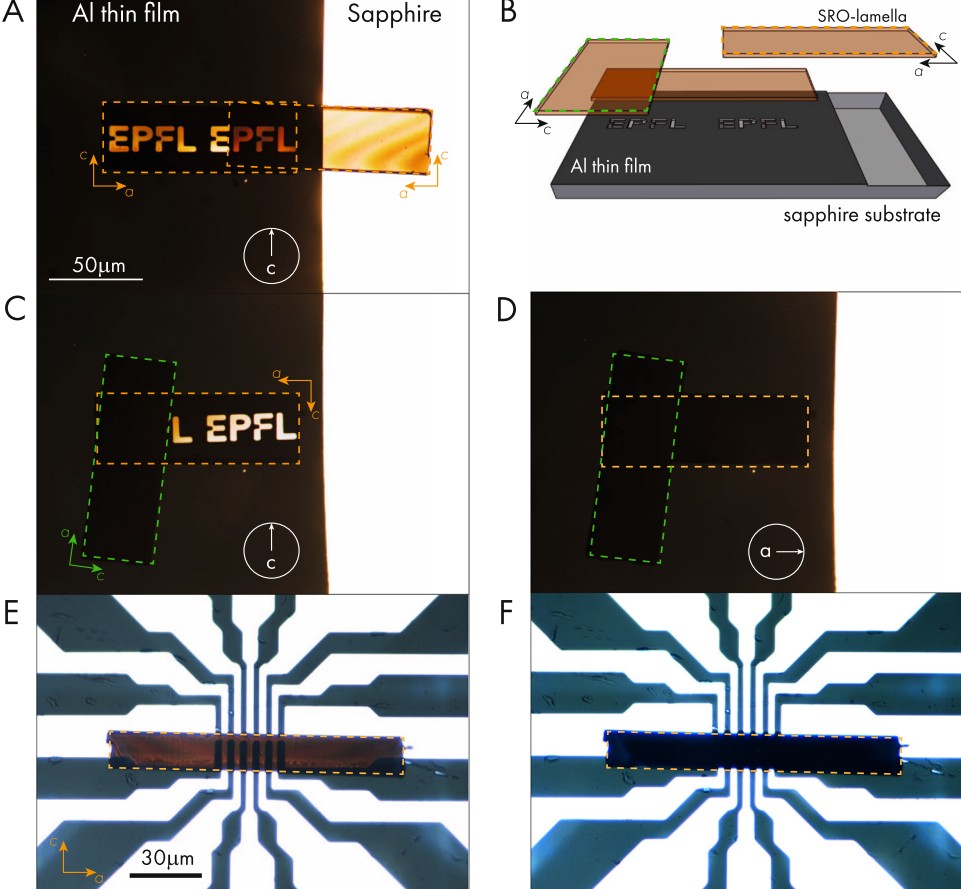

**Fig. 3 | Linear polarizing transparent conductor. A**, **C**, **D** Optical microscope images of two slabs of SRO with a thickness of 1 μm each are placed on each other with a parallel orientation. **B** Schematic of the setup used to demonstrate the linearly polarizing property of SRO. One slab is placed on a patterned Al-thin film on sapphire. A second sample can be placed either parallel (**A**) or perpendicular (**C**) to the first slab. Firstly, this demonstrates the transparency of SRO even at macroscopic thicknesses. The linear polarization can be further demonstrated by engaging a polarization filter in the microscope. The orientation of the filter is indicated in each panel. For panel **D** the filter was set to allow only a-axis polarized light to pass. The EPFL logo visible in panels **A** and **C** is absent as the sample only allows c-axis polarized light to pass. The bottom row (**E**, **F**) shows the same effect for an ac-slab of Tl2201 for c-polarized light (**E**) and a-polarized light (**F**).

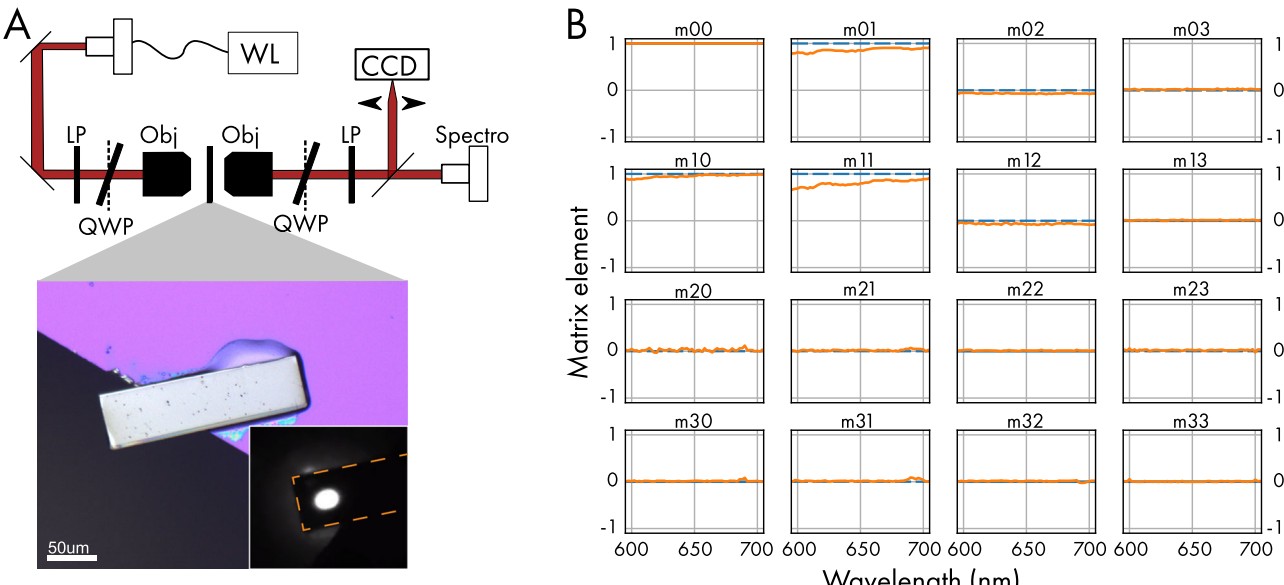

**Fig. 4 | Mueller Matrix. A** Mueller polarimetry microscope, including white light source (WL), linear polarizers (LP), motorized rotator quarter-wave plates (QWP), microscope objectives (0.55NA, 10 mm working distance, Obj), CCD camera and fiber coupled spectrometer. Micrograph of the SRO slab sample under white light and laser illumination (inset). **B** Measured wavelength-dependent Mueller matrix coefficients. The solid curves show the data while the dashed line represents the results expected for an ideal linear polarizer. The small deviation from the expected result for an ideal polariser highlights the metrological challenge.

the substrate covering the logo. The second sample was then placed either parallel or perpendicular to the first one. The orientation of the linearly polarized light used for back illumination is marked by arrows in each panel. Even when placing two slabs on top of each other, reaching macroscopic thicknesses of at least 2 µm, the samples transmit a substantial amount of light, which is most visible in Fig. 3A. Here both slabs are placed parallel to each other. With the polarization filter set to select linearly polarized light parallel to the c-axis one can see the EPFL logo shine through both devices. Rotating one sample by 90° demonstrates the polarizing properties of the quasi-2D electronic structure of SRO. While the light is still passing through the lower sample it is blocked by the top one due to the angular offset. Adjusting the polarization filter to only allow light polarized along the a-axis all light passing through the sample is blocked.

In order to demonstrate the wider prospect of this approach we prepared a 1-µm-thick ac-slab of the high-temperature superconductor $Tl_2Ba_2CuO_{6+\delta}$ (Tl2201). Tl2201 has the same crystal structure as SRO, replacing the conductive RuO-layers by CuO-layers and an almost twice as long c-axis lattice constant[13]. This structural difference is reflected in the electrical conductivity where the in-plane resistivity can be tuned from 150 to 500 µΩcm[14,15] by varying the oxygen concentration and the out-of-plane resistivity is more than 1000 times higher[16]. The Fermi surface topology of Tl2201 consists of a single quasi-two-dimensional cylinder[17]. The resulting anisotropy of the Fermi velocity $v$[18] becomes apparent in the anisotropy of the plasma frequency[19], $\omega_i^2 \propto \int \frac{v_i^2}{|v_i|} dS$, with $i$ denoting the in-plane or out-of-plane component and $dS$ an element of Fermi surface. Figure 3E, F demonstrates that the optical transmission of the single band material Tl2201 is comparable and equally polarized to that of the multiband system SRO, allowing for a simple but elegant explanation of our results by means of bandstructure properties.

For any application as a transparent conductor, the frequency response across the visible spectrum is important. To check this, we placed the device in Fig. 2 in an optical microscope with bottom illumination passing through a pinhole and polarizer. The transmitted light was collected by an objective and brought to a dispersive monochromator. The transmission spectra were taken by moving the objective between the lamella and nearby open region. We find

more than 80% of the light polarized along the crystallographic c-axis, perpendicular to the layered structure, is transmitted. For in-plane polarized light, a-axis, the transmission is strongly reduced. These observations are in perfect agreement with previous measurements of the reflectivity on ac-polished faces of bulk crystals[20,21]. It should be pointed out that this preliminary mapping showed some of the light being collected from regions not passing through the sample, due to imperfect collimation, suggesting the results in Fig. 2 slightly overestimate the transmission.

To evaluate the polarization properties of the SRO slab in more detail, we conducted broad-band Mueller polarimetry using a homebuilt polarization microscopy setup, as depicted in Fig. 4A. We find that the Mueller matrix at a wavelength of 650 nm is close to that expected for linearly polarized light in this configuration (see "Methods")[22]. Figure 4B shows the wavelength dependence of the coefficients of the Mueller matrix of the sample, yielding weak chromaticity and negligible parasitic birefringence or dichroism. To quantify this further we have determined the linear extinction ratio of the SRO sample. This parameter is often used to benchmark the performance of an optical transmitter in digital communication systems. The extinction ration is obtained by evaluating the necessary power for a logic level "1" (transmission) to "0" (absorption). For SRO we find an extinction ratio of $(3.5 \pm 0.5) \times 10^{-4}$ at 650 nm, by benchmarking against a commercial nano-particle polarizer. This very high extinction ratio is comparable to commercially available linear polarizers and hence renders a quantitative determination of the true, intrinsic extinction coefficient a metrological challenge.

The intrinsic polarizing capability of these layered oxides clearly sets them apart from other transparent conductors that are applied as thin films and possess no resistivity anisotropy within the film. An interesting novel approach to achieving transparent conductivity is the use of random metallic nanowire networks, but they also do not allow the linear polarization of light[23]. Systems that most closely resemble the structure of SRO are lithographically patterned Au nanoribbons[24], akin to microscopic wire grids. Here the 1D-structure of Au lines formed by nanosphere lithography are numerically predicted to show a polarizing effect. While the experimental results demonstrate a reduction in optical transmission, this effect is limited to a narrow

range in wavelength and limited to a 20% reduction in amplitude[24], in contrast to the results presented here.

## Discussion

Our work demonstrates the power of anisotropic conductors as transparent metals, by separating the spatial directions of high conductivity and of light propagation. In addition, *b*-oriented thin films add further functionality that isotropic thin films cannot provide. The transmitted and reflected light is naturally linearly polarized, hence eliminating the need for anti-reflective coating and the associated optical losses. In addition, the strong in-plane conductivity anisotropy can also be used for novel device concepts. Conceptually, the material resembles an atomic wire grid, highly conductive parallel wires separated by a poorly conductive matrix. Adding multiple electric contacts to such a slab that connect only to a subset of the wires would render parts of a continuous film individually addressable. The here observed broad range of wavelengths at which SRO demonstrates a high extinction ration sets it apart from commercial polarizers. A detailed comparison can be found in the supplementary information.

The main challenge towards technological implementation is the reliable and scalable synthesis of high-quality thin films which are oriented along an unfavorable growth direction while avoiding the chemically favored layer-by-layer growth. While clearly challenging, there are numerous reports of high-temperature superconductor films with this orientation, showing that this endeavor is feasible[25–28]. SRO is synthesizable by pulsed laser deposition as b-axis oriented thin films through appropriate choice of the epitaxial substrate[29,30]. Selecting an anisotropic substrate or a lower-symmetry cut of a substrate may provide the correct crystallographic template for anisotropic film growth. With regards to quality, c-axis oriented SRO films have recently been prepared by molecular beam epitaxy (MBE)[31,32] where the ability to precisely control the stoichiometry results in transport and superconducting properties rivaling those of single crystals[33]. One might envision the use of high-quality b-axis films in applications where linear polarization is desired, such as anti-glare or liquid crystal displays.

The here demonstrated concepts showcase powerful technologies that will become accessible once such microscopic control over large-area deposition of advanced materials is mastered in the future.

## Methods

### Crystal growth and orientation

Bulk single crystals of $Sr_2RuO_4$ were grown using methods as described in ref. 34. The crystal orientation was determined by X-ray Laue diffraction. For $Tl_2Ba_2CuO_{6+\delta}$, crystals were grown from self-flux[35]. The crystallographic c-axis was determined to be perpendicular to the platelet crystals using X-ray Laue diffraction.

### Preparation of slab of quantum material

Large slabs of SRO and Tl2201 were cut from bulk single crystals using a Xe-plasma focused ion beam (PFIB) system. Trenches on either side of the slab were cut using an acceleration voltage of 30 kV and a current of 2.5 μA. The slabs were subsequently undercut at 200 nA. In order to achieve a smooth surface a protection layer was deposited on the slab by co-deposition of Pt and C. Finial polished under a grazing incidence was performed at 15 nA. All slabs were prepared with the grazing incidence beam close to the crystallographic c-axis.

### Low strain setup for temperature-dependent measurements

In order to perform temperature dependence resistivity measurements a slab of SRO was transferred to a copper-grid using an in situ lift out process within the PFIB. The slab was fixed using FIB-assisted Pt-deposition. The grid and slab were cleaned using a broadband Argon

plasma and subsequently coated with 5 nm Ti and 150 nm gold using a power source at 200 W. The slab was then transferred to the center of a $100 \times 100$ μm² gold-coated silicon-nitride membrane (200 nm thick). A rectangular hole, slightly smaller than the slab, was cut prior to placing the sample. Fixing the slab above the hole was achieved by FIB-assisted Pt-deposition at 12 kV/7 nA allowing for a rigid connection with the membrane and electrical contact to the gold film. Patterning the membrane allows for a flexible connection of the sample to the silicon frame of the membrane where the sample can be electrically connected using silver wires and silver epoxy.

### Mueller matrix

A slab of SRO was mounted on the edge of a silicon frame using two component epoxy. A part of the slab was hanging free over the hole in the frame allowing for transparency measurements.

Broadband Mueller polarimetry was performed on a home-built dual rotating retarder Mueller matrix polarimeter. The setup consists of a broadband light source (Thorlabs MWWHF2), a polarization state generator (PSG), a polarization state analyser (PSA), and a fiber coupled spectrometer (Princeton Instrument Acton 750i, Roper Instrument CCD camera). The PSG consists of a fixed linear polarizer (Thorlabs LPVIS100) followed by an achromatic quarter-wave plate (Thorlabs AQWP05M-600) mounted in an automated rotation stage (Thorlabs ELL14). The PSA has the same optical elements as the PSG but in reverse order. The optical microscope is composed of two Nikon objectives (0.55NA, 10 mm working distance), allowing to probe the sample with a diffraction limited spot of ~1 μm. The sample image is recorded in transmission on a CCD camera (FLIR PointGrey).

The rotation of the fast axes of the quarter-wave plates in the PSG and PSA using the automated mounts modulates the intensity of the output beam in a way that reflects the optical polarizing properties of the sample. Transmission intensity measurements are taken for different angle combinations of the two waveplates. The Mueller matrix of the sample is then retrieved by fitting a theoretical model of the polarimeter to the data, accounting for the calibrated chromatic retardances and ellipticities of the waveplates, and taking the coefficients of the sample's Mueller matrix as free parameters.

By averaging the Mueller matrix over the visible range of the electromagnetic spectrum we obtain

$$M = \begin{pmatrix} 1 & 0.85 & -0.07 & 0.02 \\ 0.95 & 0.80 & 0.08 & 0.01 \\ -0.02 & 0.03 & 0.01 & 0.03 \\ 0.02 & 0 & 0.02 & 0 \end{pmatrix}$$

which is close to the ideal linear polarizer for which is given by

$$M = \begin{pmatrix} 1 & 1 & 0 & 0 \\ 1 & 1 & 0 & 0 \\ 0 & 0 & 0 & 0 \\ 0 & 0 & 0 & 0 \end{pmatrix}$$

The component $m_{00}$ is normalized to 1 at all wavelengths.

Another series of measurements was performed to accurately quantify the linear polarization properties of the sample and obtain its extinction ratio. To this end, the sample was placed after a reference rotating linear polarizer (Thorlabs LPVIS100) and the pair extinction ratio was measured. Under the assumption of good linear polarizers (extinction ratios close to 0), the extinction ratio of the sample is retrieved as the difference between the pair extinction ratio and the extinction ratio of the reference polarizer. The pair extinction ratio is given by the square of the deviation angle at which the transmitted intensity is equal to twice the transmitted intensity at optimal extinction. The extinction ratio of the reference polarizer is obtained by

substituting the sample by another, nominally identical linear polarizer (LPVIS100), and dividing this pair extinction ratio by 2.

**Polarization-dependent transmission in the range of visible light**
Measurements were conducted by using a WiTec alpha 300R confocal Raman microscope, using bottom illumination lamp and collecting spectra data via a fiber coupled spectrometer. A polarizer was placed between the white light lamp and the sample to control light polarization at the sample, and light was focused onto the sample through a 100× objective. The sample was mounted on a transparent sapphire substrate allowing for transmission collection, and a custom slide was designed to create an aperture just around the sample in order to ensure that only signal through the sample was collected. All spectra were repeated through the bare substrate to collect a reference. Data was obtained by taking the transmission spectra with the light polarized along the 2 in plane crystal axes. To get maximum spectral resolution a 2400 g/mm grating was used and the full spectral range was generated by stitching together all the spectra from each overlapping range.

**Reporting summary**
Further information on research design is available in the Nature Portfolio Reporting Summary linked to this article.

## Data availability

Data that support the findings of this study is deposited to Zendo with the access link: https://doi.org/10.5281/zenodo.7883517.

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

## Acknowledgements

We thank Liam Malone for help in the preparation of the Tl2201 samples. This project was supported by the European Research Council (ERC) under the European Union's Horizon 2020 research and innovation program (grant no. 715730). V.P. and K.S.B. are grateful for the primary support of the US Department of Energy, Office of Science, Office of Basic Energy Sciences under award no. DE-SC0018675. A.C. was supported by UK EPSRC grant number EP/R011141/1. N.K. is supported by a KAKENHI Grants-in-Aids for Scientific Research (Grant Nos. 18K04715, 21H01033, and 22K19093), and Core-to-Core Program (No. JPJSCCA20170002) from the Japan Society for the Promotion of Science (JSPS) and by a JST-Mirai Program (Grant No. JPMJMI18A3). Research in Dresden benefits from the environment provided by the DFG Cluster of Excellence ct.qmat (EXC 2147, project ID 390858940).

## Author contributions

A.P.M. and N.K. provided samples of $Sr_2RuO_4$. A.C. and J.R.C. provided crystals of $Tl_2Ba_2CuO_6$. C.P., C.G., and M.B. cut the slabs of S.R.O. and Tl2201. C.P. carried out the electrical transport measurements. V.P. and K.S.B. carried out the polarization measurements. M.K., T.C., M.S., P.W., and A.I. carried out measurements to determine the Müller Matrix. J.F. and S.G. studied the possibility to achieve desired thin film. C.P. and P.J.W.M. design the experiment and wrote the manuscript with input from all co-authors.

## Funding

## Competing interests

The authors declare no competing interests.
