## [Peer Review File · Nature Communications]

Layered Metals as Polarized Transparent ConductorsREVIEWER COMMENTS

Reviewer #1 (Remarks to the Author):

The manuscript by Putzke et al demonstrates the anisotropic material properties (transparency and conductivity) of layered correlated transition metal oxides Sr₂RuO₄ and Ti₂Ba₂CuO₆, and the concept for potential use as polarizing materials. The idea of using the anisotropic transmittance and reflectance induced by the free carriers for optical polarization application is interesting and innovative. However, this manuscript is lack of new discovery in materials science nor exploration of new application. The anisotropic conductivity and optical properties in layered oxides have been well known. I am not familiar with the field of materials for polarizers. To further highlight the advantage of the layered oxides as potential materials for polarizers, I would suggest the author to make full comparison and discussions with commercial available nano-particle/linear polarizer, and/or more exploration for such application.

Furthermore, a few technical issues should be addressed.

1. In Figure 1b, the conductivity for ITO used for comparison is in the range of 10²-10³ S/cm. The conductivity reported in the literature or achieved in the commercial ITO is in the range of 10⁴ S/cm or higher (Semicond. Sci. Technol. 2005, 20, S35).
2. The experimental details for measuring the in-plane and out-of plane transmissions (Figure 2C) should be provided.
3. What is the conductivity along the out-of plane direction? Assuming the conductivity is two order of magnitude lower than the in plane direction, the conductivity is still in the range of 10²-10³ S/cm. One can not expect 80% transmission in the visible light region for 400 nm thick Sr₂RuO₄, because of the d-d transition or reflection by free carriers.
4. In the introduction, the author review (discuss) the search for transparent conductive oxides and trade-off between transparency and conductivity. However, I think the main content of this manuscript of the new idea of applications of the anisotropic properties of layered oxides. I would suggest to add more review/discussions on this aspect, so as to highlight the significance of this work.
5. Minor changes. The three sub-figures (A, B, C) in Figure 2 should be captioned. Change the unit (S/m) in Figure 1b to S/cm, in order to be consistent with others in the manuscript.

Reviewer #2 (Remarks to the Author):

Putzke et al are reporting on the use of layered perovskites as polarization dependent transparent conducting oxides. These oxides show anisotropic conductivities due to the layered structure allowing for different in plane and out of plane crystal structures and builds upon their previously reported work. The idea of a polarized TCO leads to interesting applications which motivates this work for publication in Nat. Comm. After looking at the previous publications in line with this report it appears that fundamental characterization data is missing, specifically the verification of the crystal planes produced through the ion milling process. At the same time, the concept behind the paper is the mobilities, and thus carrier densities, in the different planes impact the plasmonic frequency (equ 1 and 2), yet there is no calculation or estimation of any of these values, just transmission measurements. The manuscript would be strengthened if the authors would apply the theory to either estimate the charge carrier densities, mobilities, or adsorption coefficients from the various measurements that have been taken. This is particularly true for the Ti₂201 system where the electronic measurements are not readily available like they are in the previous publications from the authors.

On a side note, the color spectrum in Figure 1C is backwards with red as the high energy range and blue for the low energy range. Also, the references need to be updated as some of them are missing information (e.g., ref 42).

Reviewer #3 (Remarks to the Author):

The authors reported the polarized transparent conductors of Sr₂RuO₄ having electrical and optical anisotropy. Although this research has novel idea, the authors should address some questions.

<Major issues>

1. Although the authors propose that modulation of plasma frequency(ω_p) makes superior transparency with high conductivity, there are merely description about plasma frequency of Sr₂RuO₄. To adjust the value of effective electron mass, dielectric constant from reference paper, calculate plasma frequency of Sr₂RuO₄.
2. While the authors show outstanding anisotropy of optical transmission, have to examine growth status of Sr₂RuO₄. It is important to check c-axis growth of Sr₂RuO₄ as well as single direction and phase.
3. The author need to confirm whether the extinction ratio of SRO(3.5×10^{-4} at 650nm) are comparable with previous reports containing bulk and thin film experiment. It is best to measure the extinction ratio of SRO film before FIB.
4. In transparent conductor research, thickness is so importance factor. Although the authors show high transmittance about 1 μ m thick SRO in figure 3, we wonder the measured range of optical microscope. Dose it fully covers visible range?

<Minor issues>

1. To describe the plasma frequency of metal, contain the Nickel in your description. There is denotes about nickel only in the figure caption.
2. In figure1 caption, there is confusion about duplicated expression about in-plane notation. In the fifth line of Figure 1 caption, there are σ_a and σ_{ab} .
3. Identify the descriptions of caption number. i.e., "a," and "(a)" are different in figure 1 and 3, respectively.
4. In figure 2 B, it is better to compare resistivity with bulk value.

Reviewer #1 (Remarks to the Author):

The manuscript by Putzke et al demonstrates the anisotropic material properties (transparency and conductivity) of layered correlated transition metal oxides Sr₂RuO₄ and Tl₂Ba₂CuO₆, and the concept for potential use as polarizing materials. The idea of using the anisotropic transmittance and reflectance induced by the free carriers for optical polarization application is interesting and innovative. However, this manuscript is lack of new discovery in materials science nor exploration of new application. The anisotropic conductivity and optical properties in layered oxides have been well known. I am not familiar with the field of materials for polarizers. To further highlight the advantage of the layered oxides as potential materials for polarizers, I would suggest the author to make full comparison and discussions with commercial available nano-particle/linear polarizer, and/or more exploration for such application.

We thank the reviewer for their positive remarks on our manuscript and suggestions on highlighting the novelty and innovation. The field of layered conductors is a long quest with ever new discoveries that broaden our understanding of a wide range of materials. While the c-axis transport of these materials has been well-studied, it was only recently that novel focused ion beam (FIB) micro machining gave us a glimpse of what lies beyond the bulk properties of these materials (C.Putzke, *et al.* Science 2020). The same holds for the well established optical properties on these materials, which have been performed in a reflection geometry. In this manuscript we demonstrate that the application of FIB leads to new opportunities in the field of layered conductors. By producing slabs of layered quantum materials, it is now possible to study optical properties of these materials not only in reflectivity but also in transmission geometry in the visible range of light. The finite size on the sub-micrometer level further enables the production of micro cavities for optical light that are impossible in macroscopic crystals.

As suggested by the reviewer we have expanded our comparison to commercially available polarizers. There is a wide variety in the market, where the prices are driven by combining high extinction ratio with a wide range in wavelength. We have summarized the properties in the supplementary information. The polarization based on the anisotropy in conductivity in the layered materials presented here is most closely related to Dichroic Polarizers. The extinction ratio in our search showed to be >1000 with values up to 100.000. The maximum extinction ratio in these systems is however very frequency dependent which is due to the artificial creation of an anisotropic conductor. A natural consequence of a crystalline stacking of conductive planes is their atomic scale. Unlike much larger structures such as polymers and nanowires, the single-atomic conductive planes are substantially below the wavelength of even blue/violet light, up to the scale of X-ray photons. Hence diffraction effects are negligible, and a flat performance up to the limits set by interband transitions is demonstrated.

For the layered conductors presented here the absence of any further interface and high homogeneity and purity of the single crystal quantum materials allow for an extinction ratio which is almost frequency independent as shown in fig. 2 and 4.

The same is true for thin film, reflective and power polarizers. The preparation of this systems is tailored for specific wave length.

We thank the reviewer for the careful and detailed review of our manuscript and will address further technical remarks below:

[supplement comparison]

1. In Figure 1b, the conductivity for ITO used for comparison is in the range of 10²-10³ S/cm. The conductivity reported in the literature or achieved in the commercial ITO is in the range of 10⁴ S/cm or higher (Semicond. Sci. Technol. 2005, 20, S35).

We thank the reviewer for bringing this to our attention and have added the relevant datapoints shown in these publications.

2. The experimental details for measuring the in-plane and out-of plane transmissions (Figure 2C) should be provided.

We have expanded the method section to include a detailed description of the setup.

3. What is the conductivity along the out-of plane direction? Assuming the conductivity is two order of magnitude lower than the in plane direction, the conductivity is still in the range of 10²-10³ S/cm. One can not expect 80% transmission in the visible light region for 400 nm thick Sr₂RuO₄, because of the d-d transition or reflection by free carriers.

The electrical conductivity of SRO in the RuO-layer is given by figure 2B to be 1.25 S/cm at room temperature. The electrical anisotropy of SRO is estimated to 200 at room temperature, which is in good agreement with room temperature values obtained in our optical transmission devices. We therefore obtain an electrical conductivity of 7 mS/cm. This low carrier mobility is in good agreement with observed optical transmission in macroscopic thicknesses.

We have expanded the discussion on the electrical conductivity and its anisotropy to improve the transition from electrical to optical properties.

4. In the introduction, the author review (discuss) the search for transparent conductive oxides and trade-off between transparency and conductivity. However, I think the main content of this manuscript of the new idea of applications of the anisotropic properties of layered oxides. I would suggest to add more review/discussions on this aspect, so as to highlight the significance of this work.

We thank the reviewer for the suggestions.

At the core of this and many other applications is the geometric separation of the propagation of photons and electrons in these layered materials. While the c-axis conductivity holds a plasma frequency below the optical spectrum, the in-plane direction shows a high electrical conductivity.

The reviewer gave an example of highly conductive, state of the art ITO above. In this a resistivity of 72 μΩ cm (Semicond. Sci. Technol. 2005, 20, S35) was achieved. This is comparable with the in-plane resistivity of our SRO slabs. However, in order to achieve a transparency of 90% the ITO film can only be 30nm thick (Japan. J. Appl. Phys. 40 L401) due to its high carrier mobility. This is more than ten times thinner than our SRO slabs. Therefore, in an application with the same cross section ITO would show an order of magnitude higher resistance and hence an order or magnitude higher thermal dissipation than our SRO slab. This difference in resistance is non-negligible in cases of high-power optical devices. These considerations appear to us as the most natural point to connect our observation to the current state of the field, as in current approaches the optical and electrical conductivities oppose each other, hence the common trade-off.

However, we fully agree that a key point is to explore further use cases of anisotropy in relevant challenges in applications. For this we also present their use as optical polarizers. As highlighted above, its wavelength independent nature is an interesting property. Yet this application is quite niche compared to transparent conductors, and much less prevalent in the field (in resonance with your own statements). We feel that shifting the focus in this direction would render our work less accessible to the non-expert.

5. Minor changes. The three sub-figures (A, B, C) in Figure 2 should be captioned. Change the unit (S/m) in Figure 1b to S/cm, in order to be consistent with others in the manuscript.

We thank the reviewer for pointing this out. We have changed the relevant figures and agree that this helps the clarity of the manuscript.

Reviewer #2 (Remarks to the Author):

Putzke et al are reporting on the use of layered perovskites as polarization dependent transparent conducting oxides. These oxides show anisotropic conductivities due to the layered structure allowing for different in plane and out of plane crystal structures and builds upon their previously reported work. The idea of a polarized TCO leads to interesting applications which motivates this work for publication in Nat. Comm. After looking at the previous publications in line with this report it appears that fundamental characterization data is missing, specifically the verification of the crystal planes produced through the ion milling process. At the same time, the concept behind the paper is the mobilities, and thus carrier densities, in the different planes impact the plasmonic frequency (equ 1 and 2), yet there is no calculation or estimation of any of these values, just transmission measurements. The manuscript would be strengthened if the authors would apply the theory to

either estimate the charge carrier densities, mobilities, or adsorption coefficients from the various measurements that have been taken. This is particularly true for the TI2201 system where the electronic measurements are not readily available like they are in the previous publications from the authors.

On a side note, the color spectrum in Figure 1C is backwards with red as the high energy range and blue for the low energy range. Also, the references need to be updated as some of them are missing information (e.g., ref 42).

We thank the referee for taking the time to review our manuscript and for the positive feedback. As suggested by the reviewer we have added estimates of the plasma frequency of SRO and TI2201 in the different crystallographic directions. For this we have applied the room temperature conductivity and the electronic anisotropy. We have further added in-plane resistivity data of TI2201 in the supplementary information. Both for SRO as well as for TI2201 the resistivity values obtained at room temperature are in good agreement with previous reports in bulk single crystals. The fabrication of small ohmic contacts in the cuprate superconductors has been done by annealing recipes which are not directly transferable. The data on TI2201 is hence taken on a geometry in which it was not possible to simultaneously probe optical and electrical conductivity.

For the crystallographic alignment of our slabs we have used single crystal XRD to establish the c-axis of the bulk single crystal which serves as our starting point. Both SRO as well as TI2201 grows a plate-like crystals with large flat surfaces of in-plane orientation. Within the FIB system we are able

to align this surface to better than $\pm 0.5^\circ$. A number of different samples have been produced and the optical transmission was confirmed for all samples. This demonstrates that within the uncertainty of the alignment with better than 1° does not lead to changes in the result but merely a deviation in the determined thickness of our slabs in the in-plane direction. This uncertainty in thickness due to the alignment is $<2\%$.

Reviewer #3 (Remarks to the Author):

The authors reported the polarized transparent conductors of Sr₂RuO₄ having electrical and optical anisotropy. Although this research has novel idea, the authors should address some questions.

1. Although the authors propose that modulation of plasma frequency(ω_p) makes superior transparency with high conductivity, there are merely description about plasma frequency of Sr₂RuO₄. To adjust the value of effective electron mass, dielectric constant from reference paper, calculate plasma frequency of Sr₂RuO₄.

We thank the referee for the suggestion to add the estimated plasma frequencies of SRO based on the resistivity anisotropy.

2. While the authors show outstanding anisotropy of optical transmission, have to examine growth status of Sr₂RuO₄. It is important to check c-axis growth of Sr₂RuO₄ as well as single direction and phase.

We agree with the referee that the crystallographic alignment of the thin slab is of high importance. In our experiment this is done by single crystal XRD prior to production of the thin slab. A polycrystalline sample or phase would produce x-ray reflections that can not be indexed.

3. The author need to confirm whether the extinction ratio of SRO(3.5×10^{-4} at 650nm) are comparable with previous reports containing bulk and thin film experiment. It is best to measure the extinction ratio of SRO film before FIB.

We thank the referee for this suggestion. Unfortunately, a direct comparison of SRO before FIB microstructuring is not possible. A transmission measurement requires a thin slab in the (*ac*)-plane which cannot be made by conventional techniques due to their tendency to cleave. To actually be able to make this slabs is a main technical advancement presented here. So far the size of bulk single crystals has only allowed measurements of reflectivity in the here presented geometry. Transmission has been probed on (*aa'*)-aligned films on platelets, however due to the high in-plane conductivity, the material's behavior in this configuration does not differ from that of a normal high carrier density metal like copper. Due to FIB microstructuring it is now possible to produce thin slabs of SRO from a bulk single crystal. Compared to conventional thin film techniques we obtain a highly oriented slab in any desired direction, which is free standing and does not require a substrate. This allows us to access crystallographic directions which have been difficult to achieve by MBE so far as SRO preferentially grows in the in-plane orientation.

4. In transparent conductor research, thickness is so importance factor. Although the authors show high transmittance about 1 μm thick SRO in figure 3, we wonder the measured range of optical microscope. Dose it fully covers visible range?

We thank the referee for raising this important point. The optical images presented in the manuscript are for illustration purpose only and were not used to extract the transmission and reflectivity. The images showcase the optical properties of SRO and Ti2201 in a crystallographic direction where thin films are currently inaccessible and FIB microstructuring offers us a new avenue to explore these properties. In order to evaluate the transmittance of SRO and its wavelength dependence in more detail we have not evaluated the optical images but used a different setup employing wavelength resolved optical transmission shown in figure 2C as well a determine the Muller matrix elements by polarometry in figure 4. We have included a detailed description of the measurements conducted in figure 2 in the methods section.

Minor Points

1. To describe the plasma frequency of metal, contain the Nickel in your description. There is denotes about nickel only in the figure caption.
2. In figure1 caption, there is confusion about duplicated expression about in-plane notation. In the fifth line of Figure 1 caption, there are σ_a and σ_b .
3. Identify the descriptions of caption number. i.e., "a," and "(a)" are different in figure 1 and 3, respectively. [
4. In figure 2 B, it is better to compare resistivity with bulk value.

We thank the reviewer for the suggestions and careful checks on our manuscript. We have changed the figure captions to be more consistent in the format.

A direct comparison of the resistivity values of SRO in this form is only valid at high temperature where the electron mean free path is smaller than the sample dimensions and holds consistent results with bulk SRO (Hussey *et al.* PRB 57, 5505). At low temperature the mean free path of SRO exceeds the sample dimension and will therefore not allow for a direct comparison with bulk properties. The temperature dependence of resistivity in SRO demonstrates that the fabrication of mesoscopic objects does not result in an increase in electrical resistivity at room temperature but leads to finite size effects that are beyond the scope of this manuscript.

REVIEWER COMMENTS

Reviewer #1 (Remarks to the Author):

The manuscript has been substantially improved in the revised version.

1. I agree that the idea that using anisotropic materials to separate the direction conductivity and transmission is a good idea. However, I would suggest to discuss/highlight the potential application aspect of this kind materials in the introduction, since this kind material is not “real” transparent conducting oxide (e.g., they can not be used as transparent electrode). I also have a general remark for the manuscript. In the introduction, the authors discuss the design principle and challenge for transparent conducting oxide materials. However, at the end, the authors explore the applications of Sr₂RuO₄ as polarizer. The authors also mention that the challenge lies in the device fabrication. I do not see the link between introduction and the exploration of Sr₂RuO₄ as polarization materials.

2. The authors compare Sr₂RuO₄ with commercially available polarizers in the supplementary information. What are the materials used for these commercial polarizers? I would also suggest to extend the comparisons and discussions on the advantages of Sr₂RuO₄ (e.g., wavelength independent extinction ratio and optical transmission) in the main manuscript.

Reviewer #3 (Remarks to the Author):

I evaluated that the authors properly answered all questions raised by me. Therefore, I recommend this manuscript for publications.

Reviewer #1 (Remarks to the Author):

The manuscript has been substantially improved in the revised version.

We thank the reviewer for the positive feedback and would like to thank the reviewer for the help in improving our manuscript. We will address further remarks below:

1. I agree that the idea that using anisotropic materials to separate the direction conductivity and transmission is a good idea. However, I would suggest to discuss/highlight the potential application aspect of this kind of materials in the introduction, since this kind of material is not “real” transparent conducting oxide (e.g., they can not be used as transparent electrode). I also have a general remark for the manuscript. In the introduction, the authors discuss the design principle and challenge for transparent conducting oxide materials. However, at the end, the authors explore the applications of Sr₂RuO₄ as a polarizer. The authors also mention that the challenge lies in the device fabrication. I do not see the link between the introduction and the exploration of Sr₂RuO₄ as polarization materials.

We thank the reviewer for the careful consistency check in the presentation of possible applications. Within this work we have demonstrated that focused ion beam machining allows us to confine SRO as well as Ti₂ZrO₁ to dimensions at which they become transparent. In particular, we were also able to measure the electrical resistivity of SRO while being transparent, hence demonstrating it as an effective transparent conductor. While we fully agree with the referee that in the present form and with the current fabrication technique these materials are of little use as actual transparent electrodes, this may well change with advances in our understanding of materials. We see this as an experimental glimpse into the functionalities these materials will hold in the future, when precise, on-demand atomic control becomes achievable in industrial applications. There is no fundamental reason why they cannot ever be used as electrodes, on the contrary, our work shows that they would perform well in actual application – once their challenging growth can be controlled.

Extreme anisotropy of the conductivity is the key principle that makes our metals transparent, which also sets them apart from all other approaches of transparent conductors which are generically isotropic planar conductors. It is exactly that property of anisotropy that introduces polarization-dependent responses which are otherwise absent in transparent conductors. Our approach and extreme polarization dependence are inseparable, and therefore the application of optical polarizer emerges naturally as a new aspect of transparent electronics.

The described experimental setup and procedure even demonstrates the possibility to produce a small number of handcrafted systems for applications. Ultimately, it is our goal that this observation sparks further development and work in this direction and we stimulate this by exploring the possibility of a different manufacturing route for SRO by using epitaxial growth, suitable for mass production and integration into technology.

2. The authors compare Sr₂RuO₄ with commercially available polarizers in the supplementary information. What are the materials used for these commercial polarizers? I would also suggest to extend the comparisons and discussions on the advantages of Sr₂RuO₄ (e.g., wavelength independent extinction ratio and optical transmission) in the main manuscript.

Based on the information available in the datasheets we were not able to obtain information of the metallic part of the polarizer. The main focus in this regard is put on the transmitting substrate

which is fused silica. A detailed comparison of SRO with commercial polarizers has been added in the supplementary information as it is a novel aspect of transparent conductors. In the main manuscript we aim to keep the balance between the three main benefits of SRO:

1. The large thickness and hence low resistance of the possible transparent contacts.
2. The highly polarized nature of the transmitted light.
3. The broader application of this concept to other materials, such as high temperature superconductors.

We believe that a detailed comparison is most relevant to the expert reader and have added a short summary of the main benefits and reference to the supplementary information into the main manuscript.

Reviewer #3 (Remarks to the Author):

I evaluated that the authors properly answered all questions raised by me. Therefore, I recommend this manuscript for publications.

We thank the reviewer for the effort in reading our manuscript again and appreciate the positive feedback.

REVIEWERS' COMMENTS

Reviewer #1 (Remarks to the Author):

The authors have addressed the comments I raised. I recommend this manuscript for publications.